# The risk of ergonomic injury across surgical specialties

**Ksenia A. Aaron**[1], **John Vaughan**[2], **Raghav Gupta**[3], **Noor-E-Seher Ali**[1], **Alicia H. Beth**[2], **Justin M. Moore**[4], **Yifei Ma**[1], **Iram Ahmad**[1], **Robert K. Jackler**[1], **Yona Vaisbuch**[1]*

**1** Department of Otolaryngology–Head and Neck Surgery; Division of Otology/Neurotology Lateral Skull Base Surgery, Stanford University School of Medicine, Stanford, California, United States of America, **2** Environmental Health & Safety Department, Stanford Health Care and Stanford University School of Medicine, Stanford, California, United States of America, **3** Rutgers New Jersey Medical School, Newark, New Jersey, United States of America, **4** Department of Neurosurgery, Boston Medical Center, Boston University, Boston, Massachusetts, United States of America

* yona_v@rmc.gov.il

**Data Availability Statement:** All relevant data are within the manuscript and its Supporting Information files.

**Funding:** The authors received no specific funding for this work.

## Abstract

## Objective

Lack of ergonomic training and poor ergonomic habits during the operation leads to musculoskeletal pain and affects the surgeon's life outside of work. The objective of the study was to evaluate the severity of ergonomic hazards in the surgical profession across a wide range of surgical subspecialties. We conducted intraoperative observations using Rapid Entire Body Assessment (REBA) score system to identify ergonomic hazards. Additionally, each of the ten surgical subspecialty departments were sent an optional 14 question survey which evaluated ergonomic practice, environmental infrastructure, and prior ergonomic training or education. A total of 91 surgeons received intraoperative observation and were evaluated on the REBA scale with a minimum score of 0 (low ergonomic risk <3) and a maximum score of 10 (high ergonomic risk 8–10). And a total of 389 surgeons received the survey and 167 (43%) surgeons responded. Of the respondents, 69.7% reported suffering from musculoskeletal pain. Furthermore, 54.9% of the surgeons reported suffering from the highest level of pain when standing during surgery, while only 14.4% experienced pain when sitting. Importantly, 47.7% stated the pain impacted their work, while 59.5% reported pain affecting quality of life outside of work. Only 23.8% of surgeons had any prior ergonomic education. Both our subjective and objective data suggest that pain and disability induced by poor ergonomics are widespread among the surgical community and confirm that surgeons rarely receive ergonomic training. Intraoperative observational findings identified that the majority of observed surgeons displayed poor posture, particularly a poor cervical angle and use of ergonomic setups, both of which increase ergonomic risk hazards. This data supports the need for a comprehensive ergonomic interventional program for the surgical team and offers potential targets for future intervention.

**Competing interests:** The authors have declared that no competing interests exist.

# Introduction

The operative tasks, which surgeons undertake every day, require not only mental sharpness, concentration, hand-eye coordination and precise execution of movement, but also minutes to hours of sustained posture with prolonged static exertion [1, 2]. While surgery is an inherently dynamic environment, where conditions change in a split-second, a surgeon, more often than not, assumes a poor, ergonomically limited postural position in order to ensure that the surgical area of interest is optimally exposed and accessed. Until recently, the medical field has exclusively focused patient welfare, *primum non nocere*, neglecting the physician's self-care and well-being [3, 4].

Ergonomics is "the concept of designing the working environment to fit the worker, instead of forcing the worker to fit the working environment" [5]. Recent studies have emphasized the hazards of improper workplace ergonomics within the surgical field. Globally, between 23–100% of surgeons across various subspecialties, report some degree of musculoskeletal (MSK) discomfort stemming from poor ergonomics during work [6]. Lack of ergonomic training and subsequent ergonomic practice during the operation leads not only to discomfort and pain but also results in fatigue, and can affect surgical speed and stamina, as well as concentration [4, 7]. Furthermore, outside of work, surgeons report that occupationally-induced MSK pain leads to disturbance in sleep, relationships, and has a negative effect on quality of life [7, 8].

While a recent meta-analysis of surgical ergonomics studies analyzed forty articles that examined subjective MSK symptoms and ergonomic outcomes through surveys across various surgical subspecialties [3], only a few studies have looked at objective body position measurements as it relates to intraoperative ergonomics. One such study objectively assessed intraoperative ergonomics using Rapid Upper Limb Assessment (RULA) and observed that 0% (0/275) of pediatric otolaryngologists were found to have a negligible level of ergonomic risk [9]. Other objective tools that are commonly used to assess ergonomics include the Ovako Working Analysis System, Posture, Activity, Tools, and Handling analysis, and the Rapid Entire Body Assessment (REBA) [8].

The REBA is a standardized observational tool developed to enable quantitative measurement of postural strain and discomfort. It does so by scoring overall ergonomics by evaluating different body segments for muscle activity caused by static, dynamic, rapidly changing or unstable postures [10]. The tool is available as an app, which enabled continuous real-time assessments and documentation of the risk of ergonomic injury. Our group previously evaluated REBA through observing Otolaryngologists at our institution and examining how objective and subjective scores correlated to ergonomic hazards [8]. Based on the success of the initial study, we expanded the project to complete a cross comparison analysis between multiple surgical subspecialties. This study aimed to explore the occupational risk to surgeons across multiple surgical subspecialties by comparing objective and subjective measures of ergonomic hazard. It set out to identify the prevalence of pain, prior ergonomic knowledge, and the influence of former ergonomic education on future OR behavior. Our manuscript is novel as it assesses both objective and subjective intraoperative ergonomic hazards and severity of MSK symptoms across multiple surgical specialties.

# Materials and methods

## Study design and participants

This was a prospective observational study of a cohort of surgeons in a tertiary hospital setting. For enrollment, our team attended Departmental meetings, where the aim of the study was explained, and surgeons were recruited at will and verbally consented for participation. For

this study, 389 surgeons were invited to participate in the study from ten surgical programs, including: Cardiovascular, General, Neurosurgery, Obstetrics and Gynecology, Ophthalmology, Orthopedic, Otolaryngology, Plastics, and Vascular. Interventional Radiologists were also included. Of the 389 surgeons invited, 167 (43%) responded to the anonymous survey. The participants consisted of attending staff, residents, and fellows. No exclusion criteria were set in place as there was an ongoing recruitment of willing participants. The study was reviewed by the Stanford University Institutional Review Board, prior to commencement, and was deemed as a quality assurance / quality improvement study.

## Survey: Subjective ergonomic hazard evaluation tool

Stanford Healthcare's Environmental Health and Safety Ergonomics Program, in collaboration with the Stanford Department of Otolaryngology-Head & Neck Surgery conducted a study of selected occupational health effects associated with the practice of surgery between 2016 to 2018. A 14-question survey evaluating self-reported (S1 Table) MSK discomfort, knowledge of ergonomics, and availability of ergonomic equipment was generated using online software (Survey Monkey, Palo Alto, California, USA) [8]. The survey also evaluated the number of years the participant was from the initiation of surgical training. The survey was distributed by e-mail to the faculty and surgical trainees across the ten surgical and interventional departments at Stanford Medical Center. The physicians were allotted one month to complete the anonymous survey. Throughout this period, one reminder e-mail was sent. At the conclusion of the survey, responses were collected and analysed.

## Rapid Entire Body Assessment (REBA)—objective ergonomic hazard evaluation tool

The objective evaluation of ergonomic injury risk was done using a validated ergonomic risk assessment tool called Rapid Entire Body Assessment (REBA) after obtaining a verbal consent [10, 11]. A complete description of REBA can be found in our previously published data and briefly presented here [8]. As recorded in the app, the neck, trunk and leg position were evaluated first and the total score for the evaluation provided at the end (Fig 1A). Upper extremities were evaluated second, specifically the upper and lower arm as well as wrist position (Fig 1B). Finally, the activity score, which accounts for the length of time in a position and the repetitiveness of activity, is given in Fig 1C. Then the final REBA score for our surgeon's evaluation was recorded and reported in our manuscript (Fig 1D). Observation of surgeons during surgery took place over several months and covered a range of postures: seated and using microscopic interventions, seated and conducting robotic surgery, standing open surgery and standing endoscopic. The surgeons were verbally consented to participate in observation by the Ergonomics team initially prior to observation and then on the day of the observation in the operating room. Observations of the participant were conducted on the availability of the Ergonomics team as well as the surgeon on any given date. Not all of the participants that completed the survey were observed in the operating room setting. The REBA scores range from 0 to 15. Increases in REBA stratifying scores indicates ergonomic injury risk and is interpreted as follows; low risk (3 or less), medium risk (4 to 7), high risk (8 to 10), and very high risk (11 and above). On average Ophthalmology surgeries were 30 minutes or less, while the remaining subspecialties had intraoperative observation done over 2 hours, at which point the observation was stopped.

## Statistical analysis

A chi-square test and Fisher's exact test were used to test the association between two categorical variables. Chi-square was used when the expected number of subjects in every cell was five

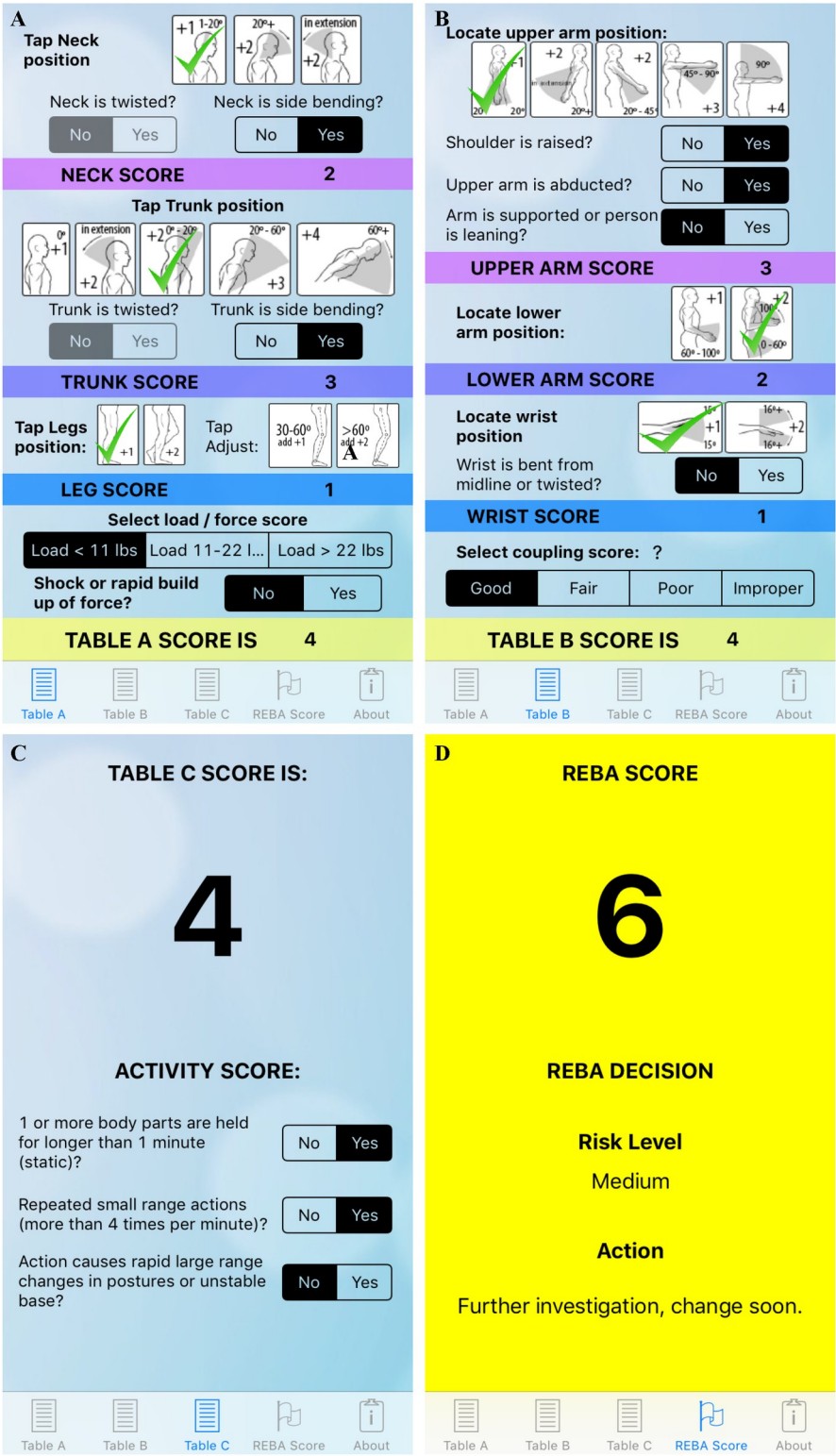

**Fig 1. REBA app for intraoperative observation of ergonomic risk assessment severity.** (A and B) REBA score for limbs as well as for the head and neck evaluation. (C) Activity score. (D) The REBA final risk level assessment score and recommendation, provided at the end of the assessment where the color represents severity risk as follows: green = low risk (3 or less), yellow = medium risk (4–7), orange = high risk (8–10), red = very-high risk (11 and above).

or more. When the expected number of subjects in any cell fell below five a Fisher's exact test was applied. To compare REBA scores between different categories of covariates a t-test was utilized. For analysis, our team used the SAS 9.4 (SAS Institute, NC) statistics software package and a p value of less than 0.05 was considered statistically significant.

## Results

Out of the 389 surgeons that received the survey, 167 (42.9%) responded across eight out of ten programs that the surveys were sent to. The surgical departments participating in the survey comprised of Otolaryngology (n = 48, 28.8%), Ophthalmology (n = 26, 15.6%), Obstetrics and Gynecology (n = 21, 12.6%), Cardiovascular (n = 15, 8.9%), Neurosurgery (n = 3, 1.8%), General, Plastics, and Vascular surgery had only one respondent (0.6%) in each and 51 (30.5%) of the surgeons did not specify their subspecialty. In our cohort, 67 of the surgeons were female (40.1%) and 100 (59.9%) were male. A third of the responders (29.0%) were in their initial five years of surgical training.

### Ergonomic knowledge and accessibility

There were 155 participants that responded to questions on ergonomics knowledge (Table 1). Of those 155, 118 (76.1%) surgeons had no prior ergonomics training while 37 had some ergonomics training prior to the study. Eight (5.2%) stated that training occurred during medical

**Table 1. Responses to the ergonomics survey questionnaire.**

| Training prior to study | N (%) |
|---|---|
| Never | 118 (76.1%) |
| Residency | 13 (8.4%) |
| Other | 13 (8.4%) |
| Medical school | 8 (5.2%) |
| Expert | 3 (1.9%) |
| **No access to ergonomic chair/stools** | |
| In operating room | 47 (28.1%) |
| In clinic | 26 (15.6%) |
| **Position most commonly taken when operating** | |
| **Standing** | |
| Open | 87 (52.7%) |
| Endoscopic | 27 (16.4%) |
| Microscopic | 5 (3%) |
| **Sitting** | |
| Microscopic | 42 (25.5%) |
| Endoscopic | 2 (1.2%) |
| Robotic | 2 (1.2%) |
| **Activity that causes the most back pain** | |
| Standing (surgery) | 72 (55%) |
| None | 29 (22.1%) |
| Sitting (surgery) | 13 (9.9%) |
| Other | 12 (9.2%) |
| Sitting (clinic) | 5 (3.8%) |
| **Significant discomfort while operating** | |
| No pain | 51 (30.5%) |
| Both the lumbar and cervical | 43 (25.8%) |

(*Continued*)

**Table 1.** (Continued)

| | |
|---|---|
| Cervical neck pain | 35 (21%) |
| Limb | 26 (15.6%) |
| Lumbar pain | 12 (7.2%) |
| **Frequency experiencing this pain** | |
| <once/week | 61 (48%) |
| 1–2 times | 41 (32.3%) |
| > = 3 times | 23 (18.1%) |
| Continuously | 2 (1.6%) |
| **Discomfort/pain affected work** | |
| Not at all | 67 (52.3%) |
| Mildly | 42 (32.8%) |
| Moderately | 17 (13.3%) |
| Severely | 2 (1.6%) |
| **Discomfort/pain affected outside work** | |
| Not at all | 53 (40.5%) |
| Mildly | 49 (37.4%) |
| Moderately | 26 (19.9%) |
| Severely | 3 (2.3%) |
| **Severity of MSK pain, mean (± S.D.)** | 2.9 (± 2.0) |

school, 13 (8.4%) received it while in residency, 3 (1.9%) obtained it from an expert consultation, while another 13 (8.4%) reported other means of receiving training.

In regard to ergonomic accessibility, out of 167 surgeons, 141 (84.4%) stated that they did not have access to ergonomic chair/stools in their clinic. Furthermore,120 (71.8%) surgeons reported no access to this equipment in the operating room.

## Subjective ergonomic hazard assessment

As shown in Table 1, across the 167 responders, the most frequently assumed position during surgery was standing, especially during open surgical procedures (n = 87, 52.7%). The standing position was also the most common position leading to back pain (n = 72, 53.9%). A total of 69.4% of responders reported significant discomfort while operating, with the most common location of discomfort occurring in both the lumbar and cervical regions (n = 43, 25.7%). Overall, neck pain was the single most common area of pain reported (n = 35, 20.9%-cervical pain only). A third of the participants (n = 51, 30.5%) reported no significant pain during surgery. More than half of the surgeons reported experiencing pain at least once a week (51.7%), with 20.0% of the responders indicating experiencing pain more than three times per week or continuously. Using a numerical pain scale with 0 being no pain and 10 reflecting the most severe pain, the average severity of MSK pain described either during surgery or in clinic was 2.9 across subspecialties, reflecting an underlying level of discomfort.

When comparing the years from initiation of surgical training, and levels of MSK pain (Table 2), 87.7% of surgeons with less than five years of training, (43 out of 49), reported experiencing pain. For those that had more than five years of training the percentage was still high but was similar across groups: 22 out of 33 (66.7%) in those with 5–10 years, in 21 out of 36 (58.3%) from the group with 11–20 years, in 23 out of 37 (62.2%) in those with 21–30 years of training, and in 7 out of 12 (58.3%) in those with greater than 30 years of training. The overall difference is statistically significant (p = 0.02).

**Table 2. Correlation between MSK pain and years from initiating surgical training.**

| Years in surgical training | Experiencing MSK pain |
|---|---|
| <5 years | 43 (87.8%) |
| 5–10 years | 22 (66.7%) |
| 11–20 years | 21 (58.3%) |
| 21–30 years | 23 (62.2%) |
| >30 years | 7 (58.3%) |

Surgeons were further asked whether the pain affected their ability to work or perform activities outside of work. Of those that responded, 61 (47.6%) stated that pain affected their work (32.8% mildly, 13.2% moderately, and 1.6% severely), with 39 participants choosing not to respond. More surgeons indicated that pain impacted their life outside of work (n = 78, 59.5%). They reported either mild 49 (37.4%), moderate 26 (19.8%), or a severe 3 (2.3%) effect on outside activities. The rest of the responding surgeons 53 (40.5%) reported no affect, with 36 participants not responding to this question (See Table 1).

When comparing each subspecialty, neurosurgery, general, plastics, and vascular surgery had the highest number of surgeons reporting pain (100.0%). This should be interpreted cautiously as there were very few responders in each of those subspecialties (n = 3, 1, 1, 1 respectively) (Table 3). In the rest of the subspecialties, 60.0–72.9% of the surgeons described pain. Otolaryngologists (n = 15, 31.3%) and ophthalmologists (n = 8, 20.7%) noted experiencing discomfort in the cervical neck region while operating. Cardiovascular surgeons and surgeons who did not otherwise state their surgical subspecialty, reported experiencing both cervical and lumbar pain during surgery (n = 5, 33.3% and n = 14, 27.4% respectively). Obstetrics and gynecology surgeons most often experienced limb pain while operating (n = 7, 33.3%). However, the overall difference doesn't reach statistical significance (p = 0.09).

## Objective intraoperative ergonomic hazard assessment

The REBA scores were generated during intraoperative observations to evaluate the surgeon's risk of injury. A total of 91 surgeons were intraoperatively observed by REBA evaluations. The lowest REBA score for a standing surgery was 1 with the highest being 10 (n = 71, mean 6.65, S.D ±1.84). The lowest observed score for sitting operation was 0 with the highest being a 9 (n = 20, mean 3.35, S.D. ±2.39). The average difference in REBA scores for standing surgery were significantly higher than those for sitting (6.65 vs. 3.35, p<0.0001). Intraoperative

**Table 3. Significant subjective reported discomfort while operating across subspecialties.**

| | Pain location N (%) | | | | | |
|---|---|---|---|---|---|---|
| Surgical Specialty | Cervical | Lumbar | Both | Limbs | No pain | Total |
| Cardiac | 2 (13.3%) | 0 (0%) | 5 (33.3%) | 2 (13.3%) | 6 (40.0%) | 15 |
| General | 0 (0%) | 1 (100.0%) | 0 (0%) | 0 (0%) | 0 (0%) | 1 |
| Neurosurgery | 1 (33.3%) | 0 (0%) | 2 (66.7%) | 0 (0%) | 0 (0%) | 3 |
| OB/Gyn | 2 (9.5%) | 2 (9.5%) | 2 (9.5%) | 7 (33.3%) | 8 (38.2%) | 21 |
| Ophthalmology | 8 (30.8%) | 0 (0%) | 7 (26.9%) | 2 (7.7%) | 9 (34.6%) | 26 |
| Otolaryngology | 15 (31.3%) | 3 (6.2%) | 11 (22.9%) | 6 (12.5%) | 13 (27.1%) | 48 |
| Plastic | 0 (0%) | 0 (0%) | 1 (100.0%) | 0 (0%) | 0 (0%) | 1 |
| Vascular | 0 (0%) | 0 (0%) | 1 (100.0%) | 0 (0%) | 0 (0%) | 1 |
| Not reported | 7 (13.7%) | 6 (11.8%) | 14 (27.4%) | 9 (17.7%) | 15 (29.4%) | 51 |
| **Total** | **35 (21.0%)** | **12 (7.2%)** | **43 (25.7%)** | **26 (15.6%)** | **51 (30.5%)** | **167** |

**Table 4. Intraoperative objective REBA observation scores across subspecialties.**

| Surgical Specialty | N | Min | Max | Mean (± S.D.) |
|---|---|---|---|---|
| Cardiac | 6 | 6 | 8 | 7.0 (± 0.9) |
| General | 9 | 3 | 9 | 7.0 (± 1.7) |
| Interventional Rad. | 5 | 4 | 7 | 4.8 (± 1.3) |
| Neurosurgery | 7 | 6 | 10 | 8.6 (± 1.3) |
| OB/Gyn | 5 | 4 | 7 | 5.6 (± 1.1) |
| Ophthalmology | 5 | 0 | 0 | 0.0 (± 0.0) |
| Orthopedics | 6 | 7 | 9 | 7.8 (± 0.8) |
| Otolaryngology | 38 | 1 | 9 | 5.4 (± 2.1) |
| Plastics | 5 | 7 | 9 | 7.4 (± 0.9) |
| Vascular | 5 | 6 | 8 | 6.8 (± 0.8) |

observations across subspecialties are summarized in Table 4. Ophthalmology had the lowest mean observed REBA score of 0 (S.D. 0, n = 5), while neurosurgery had the highest at 8.57 (S. D. 1.27, n = 7), corresponding to low and high ergonomic risk, respectively. The remaining subspecialties fell into the medium intraoperative ergonomic risk hazard with mean scores varying from 4.80 for interventional radiology (S.D. 1.30, n = 5) to 7.83 for orthopedic surgery (S.D. 0.75, n = 6).

## Discussion

To our knowledge, our study is unique; after a comprehensive search in the English literature, we were unable to find a study that compared both objective and subjective correlation data of MSK symptoms and intraoperative ergonomic hazard across multiple surgical subspecialties. The MSK pain and injury suffered by surgeons is a frequent occurrence particularly in the context of prolonged procedures which require the surgeon to operate in a static posture for an extended period of time. This discomfort and the impact from poorly adapted ergonomics in surgical practice was validated in our paper using both objective and subjective assessments.

### Poor knowledge of ergonomics and lack of access to ergonomically favorable furniture

Previous studies, examining knowledge of ergonomics have found that a third or less of surveyed surgeons have any prior knowledge or training or proper OR ergonomics. A large survey study among the North American otolaryngologist residents and facial plastic surgeons showed that only 33.2% (n = 125) of participants had prior knowledge of ergonomic principles [12]. In line with this and other studies, our surgeon population had little knowledge of ergonomics (23.9%) [12–14]. A significant factor that contributes to poor MSK outcomes is the lack of awareness of ergonomics within the medical and more specifically, the surgical field. While there is a substantial percentage of surgeons that report MSK pain directly attributed to poor surgeon body position during surgery, a surprising 55–99% of surgeons, across multiple subspecialties, report having no prior knowledge of ergonomics principles [15].

Further compounding this issue, was that only 15.6% of the surgeons in our study had access to ergonomically favorable furniture in the office and only 28.1% reported having access to it in the operating room. In the US, the majority of studies focusing on occupational ergonomics have centered around office space and computer use [3]. An entire industry has developed providing equipment designed to reduce the risk of injury from using a computer, while

the surgical field appears to be lagging behind. The main feature of such equipment is adjustability, so that the overall workstation fits the user's stature and posture, whether the individual is sitting or standing. Adjustable chairs and stools, instrument tables, the operating table itself, and lights were noted in the ORs surveyed in this study. Some, but rarely all, of these items were adjusted to the attending physician's stature. However, the number of adjustable features varied, and many surgeons were unaware of the full range of adjustable features, while some felt these features didn't make much difference. They also did not have the training in ergonomics that would allow appropriate risk assessment and a consideration of the cost and benefit of not using the equipment to full advantage.

## Musculoskeletal discomfort as reported across surgical subspecialties and observed in the operating room

Given that this study was done across different surgical specialties, each using specialty-specific surgical techniques, the level of ergonomic hazard was reflected both in the questionnaire and the objective REBA scores. Standing was the most common position leading to back pain (n = 72, 53.9%). However, the trend of MSK discomfort across specialties was the same, with surgeons experiencing both lumbar and cervical (n = 43, 25.7%) pain, with cervical being the most common site (n = 35, 21.0%). This is in line with a recently published meta-analysis that examined a total of 5152 surveyed surgeons looking at reported pain across multiple surgical subspecialties and finding that back (50%), neck (48%), and arm or shoulder (43%) pain was the most common [3].

As with other ergonomic studies of MSK symptoms in surgeons, we found that pain was associated with forward neck flexion, which is greatest when the surgeon is standing. This is easily understood from both observation and the REBA scores. The surgeon's head is bent forward with the neck angle between 20 and 60 degrees, which translates to an increase in forces on the cervical spine, from 10 to 12 lbs. in the neutral position, to 60 lbs. in the flexed position [16]. Furthermore, this posture may be held for several minutes and, in rare cases, hours. Holding the head in this position, together with other static postures, places the surgeon at considerable risk of injury. Muscles become exhausted more quickly in static postures. The pressure of completing the task often causes the surgeon to continue through the fatigue. This strain can lead to acute and chronic MSK injury.

There is a clear trend that surgeries with longer OR times have increased ergonomic risk. The issue of sitting and/or standing while performing work tasks has been investigated extensively in recent years, primarily focused on postures for computer use [15]. We noted during our REBA observation study that many surgeries, done from a seated position, could easily adapt good ergonomic postures. This was often true for both open and microscopic procedures in neurosurgery, otological surgery, and ophthalmic surgery. While robotic assisted surgery clearly has superior ergonomics, further improvements could be made according to our observations. The foot pedal location was fixed on the console making it difficult for some to reach the pedals. Additional risks observed while using robotic consoles included surgeons maintaining a sustained forward reach posture while maneuvering the controls and not resting the controls close to their core. Furthermore, prior to performing the surgeries, the prep phase, albeit short in comparison, was often performed with poor ergonomics. Overall, sitting was better than standing from an ergonomic perspective, across the specialties. The standing position alone, on average, added up to 3 levels of risk (e.g., REBA score increased from 4 to 7) over seated REBA scores. Whether standing or sitting, micro breaks of two minutes, every 20–40 minutes, appear to be a reasonable addition to integrate for most surgical procedures [17].

## Poor ergonomics starts early in surgeons' career

In our analysis, surgeons in their early years of training had a significantly higher number of MSK complaints when compared to their senior colleagues. This is in line with a recent study in surgical residents which showed that 82.3% of residents experience some form of MSK pain, and16.3% of residents report having to scrub out of ongoing operating procedures or missing work due to their symptoms [18]. It can also be speculated that the general culture of surgeons leads to a bias toward under-reporting their own discomfort [19]. It is also possible that senior surgeons represent a form of the "Health worker survivor effect", whereby those who have significant pain from operating have moved into other roles (e.g. administration, research or early retirement), while the healthier surgeons continue to operate. Even so, nearly 60% of surgeons in our study across all years of expertise experienced MSK pain, with more than half experiencing pain at least once a week (51.7%). Another explanation for higher rate of MSK symptoms among residents is that the ergonomic settings are usually adjusted to the senior surgeon's specification, while the junior surgeon is expected to adapt. Overall, these numbers are high and alarming. MSK pain and poor ergonomics are also problematic for those considering surgery. A recent study conducted on medical students pursuing surgery found that 73% reported experiencing MSK discomfort during their rotation [20]. Over a third of the students were deterred from the surgical field, especially when the student was informed on the risk of MSK injury in surgeons [20].

## The current roadblocks and the future of ergonomics in surgery

With the recent focus on improving physician health, limiting surgical training hours, minimizing burnout and generally improving physician work-life balance, physicians are starting to look for ways to improve everyday ergonomics–(although these efforts still lag far behind those in other non-medical fields). Despite this increase in awareness of the importance of ergonomics, it is currently insufficiently incorporated into everyday surgical practice. This is partially an artifact of traditional surgical culture, in which surgeons were trained to ignore bodily discomfort and self-care. It is also partially attributed to a general OR culture of resistance to change [21, 22]. This has created a reality in which MSK injuries are underreported, even while the majority of surgeons endure occupationally induced MSK pain [19].

Improvements in surgical ergonomics can potentially alleviate risk factors that lead to MSK injuries and thus enhance the surgeon's productivity and performance, reduce time off work, prolong surgical careers and ultimately improve patient care [21, 22]. While surgical ergonomics initially gained some attention in the early 1990s, there appear to have been few improvements in practice [23]. With increasingly strong evidence of widespread occupational injury among surgeons, little research has been conducted into methods of remediation. Given the severity of the problem, there is a clear need for guideline development followed by validated and implementable interventional programs. We are calling for intervention while taking into account the cultural hurdle of making changes in the surgical profession. One could consider intervention as early as medical school.

We acknowledge that certain aspects of our study, such as subjective MSK symptoms and outcomes, have been studied in other medical specialties [3]. We further recognize that although limited, the REBA methodology is used as an initial posture analysis. In order to evaluate the actual MSK load of the surgeon during the operation, other tools such as surface electromyography (to be able to estimate dynamic force or fatigue during a tasks) [24], simulation-based ergonomics training curriculum [25], and other instrumented and assessment tools, would be more meaningful—especially as it relates to evaluating the underlying MSK disorders in this surgical cohort. An additional limitation is not having an interpersonal

correlation between reported symptoms on the survey and intraoperative REBA observations, due to the anonymous nature of the survey. Thus, conclusions on an individual level between the subjective and objective measures could not be made. Nevertheless, our study is unique as we tried to examine these correlations across a comprehensive list of surgical fields. Moreover, although there are limitations to our study, the ultimate benefits of intervention might only be seen in the future—after years of improved awareness and implementation of ergonomics have been adopted within the medical culture. We believe there is enough of evidence to support the next stage of motivating cross-institutional interventional programs in prevention and ergonomic training. In doing so, our ultimate goal is to both improve day to day complaints, and in the long run promote healthy medical careers and improve surgeons' quality of life. Future studies specifically looking at these parameters should be undertaken to correlate the relationship across multiple surgical fields. Overall, our results highlight the challenges as observed objectively and reported subjectively that arise from poor ergonomics in the operating room and we hope that this information will further provide insight for potential future targets for intervention.

## Conclusion

Our data suggests that pain and disability as a result of poor ergonomics are widespread across surgical specialties and confirms that surgeons rarely receive implementable ergonomic training in the context of surgery. Additionally, intraoperative observational findings identified that the majority of surgeons display poor posture, particularly a poor cervical angle, which leads to increased ergonomic risk hazard and is associated with subjective cervical pain. With the low rate of ergonomic awareness, lack of access to ergonomic equipment and furniture, and high rates of MSK complaints across different stages of the surgeon's work and outside life, the need for ergonomics education is imperative.

## Supporting information

**S1 Table.**
(DOCX)

## Acknowledgments

We would like to acknowledge Michael Piekry for his for constructive criticism of the manuscript.

## Author Contributions

**Conceptualization:** John Vaughan, Raghav Gupta, Alicia H. Beth, Justin M. Moore, Robert K. Jackler, Yona Vaisbuch.

**Data curation:** Ksenia A. Aaron, John Vaughan, Raghav Gupta, Alicia H. Beth, Justin M. Moore.

**Formal analysis:** Ksenia A. Aaron, John Vaughan, Raghav Gupta, Noor-E-Seher Ali, Alicia H. Beth, Justin M. Moore, Yifei Ma, Yona Vaisbuch.

**Investigation:** Yona Vaisbuch.

**Methodology:** Ksenia A. Aaron, John Vaughan, Alicia H. Beth, Justin M. Moore, Yona Vaisbuch.

**Supervision:** Ksenia A. Aaron, John Vaughan, Justin M. Moore, Robert K. Jackler, Yona Vaisbuch.

**Validation:** John Vaughan, Noor-E-Seher Ali, Yifei Ma, Iram Ahmad.

**Writing – original draft:** Ksenia A. Aaron, Yona Vaisbuch.

**Writing – review & editing:** John Vaughan, Raghav Gupta, Noor-E-Seher Ali, Alicia H. Beth, Justin M. Moore, Yifei Ma, Iram Ahmad, Robert K. Jackler, Yona Vaisbuch.

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
