## [Decision Letter · Decision Letter 0]

23 Jul 2020

PONE-D-20-13337

The Risk of Ergonomic Injury Across Surgical Specialties

PLOS ONE

Dear Dr. Vaisbuch,

Thank you for submitting your manuscript to PLOS ONE. After careful consideration, we feel that it has merit but does not fully meet PLOS ONE’s publication criteria as it currently stands. Therefore, we invite you to submit a revised version of the manuscript that addresses the points raised during the review process.

We look forward to receiving your revised manuscript.

Kind regards,

Matias Noll, Ph.D

Academic Editor

PLOS ONE

Journal Requirements:

2. Please provide additional details regarding participant consent. In the ethics statement in the Methods and online submission information, please ensure that you have specified (1) whether consent was informed and (2) what type you obtained (for instance, written or verbal, and if verbal, how it was documented and witnessed). If your study included any authors of this manuscript, please indicate this. If the need for consent was waived by the ethics committee, please include this information.

Reviewers' comments:

Reviewer's Responses to Questions

**Comments to the Author**

1. Is the manuscript technically sound, and do the data support the conclusions?

Reviewer #1: Yes

Reviewer #2: Yes

Reviewer #3: Yes

Reviewer #4: Partly

2. Has the statistical analysis been performed appropriately and rigorously? 

Reviewer #1: No

Reviewer #2: Yes

Reviewer #3: Yes

Reviewer #4: No

3. Have the authors made all data underlying the findings in their manuscript fully available?

Reviewer #1: Yes

Reviewer #2: Yes

Reviewer #3: Yes

Reviewer #4: No

4. Is the manuscript presented in an intelligible fashion and written in standard English?

Reviewer #1: Yes

Reviewer #2: Yes

Reviewer #3: No

Reviewer #4: Yes

5. Review Comments to the Author

Reviewer #1: Generally speaking, the paper is relatively well-written with some lacks in introduction/discussions and the way data collection is performed. There is some point which I want to raise, which was mentioned below.

- The authors should declare what are the specific purposes of the paper first. Additional motivation should be included in the introduction to support the importance of this study compared to those already published.

- The article in the current format is more like an academic report or project, although it has been carried out among one of the highly risk groups in terms of ergonomics. This is because, there are several better methods than REBA that provide better and more accurate results. The REBA method is used today for an initial posture analysis among the target population, while for valid scientific studies other methods are mainly used (the author can view posture analysis in articles after 2015). Even, more importantly, evaluating the actual load while performing the different tasks, using EMG, instrumented tools or simulation would probably be more meaningful, especially to link with musculoskeletal disorders. Authors may elaborate on these points in the introduction/discussion section to reinforce the significance of their study.

- The author should make more persuasive explanation and significant discussion.

- Please check all the references to see whether they are formatted correctly or not.

However, it is my observation that the paper is unacceptable for publication.

Reviewer #2: This manuscript adds some unique understand of the risk of ergonomic injury across surgical specialties. I only have some minor comments.

In your method, it is not clear if the experience the key factor of ergonomic included in the study. And it’s also not clear if the participants were always willing to be observed.

Reviewer #3: All in all this is an exciting article with a lot of potential. But it was very difficult to follow the red thread. This was partly due to the language in which, for example, many literal and structural repetitions took place (e.g. lines 111 - 116 where the sentences are just strung together). Secondly, this is due to the presentation (many values that are only displayed one behind the other without being explained, the jumping from percentages and whole values). At the same time, the research design was presented in a way that was not entirely comprehensible.

Overall, the article should be linguistically revised. Also, the introduction and the methodological part should gain in depth, e.g. in the form of additional sources and the presentation of the research designs (was the questionnaire qualitative or quantitative?)

Concrete improvements:

Lines 49-55: Specify sources for the claim

Line 56: Give correct definition of Ergonomics including source

Line 65-78: specify further. Which studies were conducted? (including sources) What is the specific research question to be clarified in this paper? Why is it exciting?

- Line 82 - 86: Unclear what 389 refers to. Did all persons participate or were they only invited?

- Headings: by Methods: confusing, because as a reader you do not directly understand whether you are with the theoretical background, the statistics or the results.

- Study design and evaluation tool: What is the concrete study design? What steps were taken? What exactly was measured? How was the allocation made? Not clear

- relate results more strongly to the research question / hypothesis. How do the results of the subchapters help to answer the question?

- Numbers and numerical values: please check what the scientific standards are. The representation in this form is not correct.

- Discussion: indicate exact studies that have already been conducted and compare your own work with them. "previous studies" is too unspecific

Reviewer #4: The paper combines three aspects of ergonomics: musculoskeletal load and risk of developing musculoskeletal disorders, knowledge regarding ergonomics and musculoskeletal pain. Idea of analysis relationship between those three is interesting and worth exploration. However, the paper teats those areas carelessly. My main objections refer to:

Authors aim of the study as: “To evaluate the relationship between presence of postural related musculoskeletal discomfort with the level of ergonomics training and the intraoperative ergonomic practice across ten surgical subspecialties” – such relationship has not been highlighted. There has been presented results in different areas.

• There is lack of description of the tool used, especially in reference to the questionnaire. Even if it has been published, few basic informations would be helpful.

• Statistical Analysis subsection informs that Chi-square test is used when the number of subjects in every cell was five or more and when the number of subjects in any cell fell below five a Fisher's exact test was applied. On what basis was accepted such rule. A chi-square is a parametric test and to apply it specific requirements must be fulfilled. It needs to be described for what purposes were those tests applied.

• There is a lot of data presented in the text. It is difficult to follow. What is the reason that Authors have not presented results of their study on figures or in tables.

• What about results of statistical analysis? It is difficult to find them.

• Application of REBA requires specific procedures in three main steps: allocation of codes to each of parts of each of body posture, assessment of codes of two groups and in the and assessment of overall load. Presentation of only the final results deprives reader of interesting information. Attitude presented in this paper can be acceptable only if in Appendices are presented results of the three steps including illustration of each posture.

• It is difficult to find on which results Authors base their conclusions.

6. PLOS authors have the option to publish the peer review history of their article (what does this mean?). If published, this will include your full peer review and any attached files.

Reviewer #1: No

Reviewer #2: No

Reviewer #3: No

Reviewer #4: No

---

## [Author Response · Author response to Decision Letter 0]

29 Sep 2020

Dear PLOS Editorial and Reviewer Committee,

Thank you very much for taking the time to read our manuscript, and for your thoughtful and insightful feedback. Please see below for an itemized, point-by-point response to each of your comments. All changes in the manuscript are highlighted. We hope you find our revised version meeting the high standards of PLOS in consideration for publication.

Reviewer #1:

Generally speaking, the paper is relatively well-written with some lacks in introduction/ discussions and the way data collection is performed. There is some point which I want to raise, which was mentioned below.

1. The authors should declare what are the specific purposes of the paper first. Additional motivation should be included in the introduction to support the importance of this study compared to those already published.

Dear Reviewer, thank you for your kind comment and for this suggestion. The last paragraph in the introduction now covers the purpose and motivation of this study.

Lines 84-89 This study aimed to explore the occupational risk to surgeons across multiple surgical subspecialties by comparing objective and subjective measures of ergonomic hazard. It set out to identify the prevalence of pain, prior ergonomic knowledge, and the influence of former ergonomic education on future OR behavior. Our manuscript is unique as it is the first to assess objective and subjective intraoperative ergonomic hazards and severity of MSK symptoms across multiple surgical specialties. 

2. The article in the current format is more like an academic report or project, although it has been carried out among one of the highly risk groups in terms of ergonomics. This is because, there are several better methods than REBA that provide better and more accurate results. The REBA method is used today for an initial posture analysis among the target population, while for valid scientific studies other methods are mainly used (the author can view posture analysis in articles after 2015). Even, more importantly, evaluating the actual load while performing the different tasks, using EMG, instrumented tools or simulation would probably be more meaningful, especially to link with musculoskeletal disorders. Authors may elaborate on these points in the introduction/discussion section to reinforce the significance of their study.

Thank you for this insightful comment. Our team used REBA as part of our methods as that is what we had. This is the third article from our research that expanded to encompass multiple surgical subspecialties using the tools tested in prior smaller studies. We therefore wanted to continue with the data gathering according to the project design. To address that as part of the manuscript and make it more clear we have inserted within our introduction the following paragraph. As well as acknowledged it in the limitations section

Lines 67-75 While a recent meta-analysis of surgical ergonomics studies analyzed forty articles that examined subjective MSK symptoms and ergonomic outcomes through surveys across various surgical subspecialties, [3] only a few studies have looked at objective body position measurements as it relates to intraoperative ergonomics. One such study objectively assessed intraoperative ergonomics using Rapid Upper Limb Assessment (RULA) and found 0% (0/275) of pediatric otolaryngologists were found to have a negligible level of ergonomic risk. [9] Other objective tools that are commonly used to assess ergonomics include the Ovako Working Analysis System, Posture, Activity, Tools, and Handling analysis, and the Rapid Entire Body Assessment (REBA). [8]

Lines 345-349 We further recognize that although limited, the REBA methodology is used as an initial posture analysis. In order to evaluate the actual MSK load of the surgeon during the operation, tools such as surface electromyography, simulations, and other instrumented tools, would be more meaningful - especially as it relates to evaluating the underlying MSK disorders in this surgical cohort. 

3. The author should make more persuasive explanation and significant discussion.

Thank you for this point, we have made additional changes and have both modified our introduction to make it more clear of our significance as well as enhanced discussion by including limitations as well as providing a more modified discussion and anticipate that these will be more persuasive and detailed

As stated above we have included the following paragraph in the introduction:

Lines 84-89 This study aimed to explore the occupational risk to surgeons across multiple surgical subspecialties by comparing objective and subjective measures of ergonomic hazard. It set out to identify the prevalence of pain, prior ergonomic knowledge, and the influence of former ergonomic education on future OR behavior. Our manuscript is unique as it is the first to assess objective and subjective intraoperative ergonomic hazards and severity of MSK symptoms across multiple surgical specialties. 

Further, as part of the limitation and succinct discussion summarizing significance of this study we have added the following to the discussion:

Lines 344-362 We acknowledge the limitation that certain separate aspects of our methodology have been previously examined in literature. We further recognize that although limited, the REBA methodology is used as an initial posture analysis. In order to evaluate the actual MSK load of the surgeon during the operation, tools such as surface electromyography, simulations, and other instrumented tools, would be more meaningful - especially as it relates to evaluating the underlying MSK disorders in this surgical cohort. An additional limitation is not having an interpersonal correlation, due to the anonymous nature of the survey. Thus, conclusions on an individual level between the subjective and objective measures could not be made. However, our study is unique as we tried to examine these correlations across a comprehensive list of surgical fields. Moreover, although there are limitations to our study, the ultimate benefits of intervention might only be seen in the future - after years of improved awareness and implementation of ergonomics have been adopted within the medical culture. Nevertheless, we believe there is enough evidence to support the next stage of motivating cross-institutional interventional programs in prevention and ergonomic training. In doing so, our ultimate goal is to both improve day to day complaints, and in the long run promote medical careers and improve surgeons’ quality of life. Future studies specifically looking at these parameters should be undertaken to correlate the relationship across multiple surgical fields. Overall, our results highlight the challenges as observed objectively and reported subjectively that arise from poor ergonomics in the operating room and we hope that this information will further provide insight for potential future targets for intervention. 

4. Please check all the references to see whether they are formatted correctly or not.

The references have been checked and our formatted according to PLOS submissions guidelines, thank you for pointing it out.

Reviewer #2: 

1. This manuscript adds some unique understand of the risk of ergonomic injury across surgical specialties. I only have some minor comments.

In your method, it is not clear if the experience the key factor of ergonomic included in the study. And it’s also not clear if the participants were always willing to be observed.

Dear Reviewer, thank you for your kind comment and for your suggestions. We are unsure of what exactly is meant by “if the experience the key factor of ergonomic included in the study” thus we cannot comment on this. To address the second statement the following has been added to the methods:

Lines 111-112 The survey also evaluated the number of years the participant was from the initiation of surgical training.

Lines 131-135 The surgeons were verbally consented to participate in observation by the Ergonomics team initially prior to observation and then on the day of the observation in the operating room. Observations of the participant were conducted on the availability of the Ergonomics team as well as the surgeon on any given date. Not all of the participants that completed the survey were observed in the operating room setting

Reviewer #3: 

All in all this is an exciting article with a lot of potential. But it was very difficult to follow the red thread. This was partly due to the language in which, for example, many literal and structural repetitions took place (e.g. lines 111 - 116 where the sentences are just strung together). Secondly, this is due to the presentation (many values that are only displayed one behind the other without being explained, the jumping from percentages and whole values). At the same time, the research design was presented in a way that was not entirely comprehensible.

Overall, the article should be linguistically revised. Also, the introduction and the 

methodological part should gain in depth, e.g. in the form of additional sources and the presentation of the research designs (was the questionnaire qualitative or quantitative?)

Dear Reviewer, thank you for your astute and thorough comments/corrections. We had thoroughly re-reviewed our manuscript as well as had a linguist review the manuscript for language and grammar and anticipate that all of the corrections made will improve readability and flow of it. To address the methodology and research design, while we initially chose to present only two tables for all of the numerical values presented in the methods, we understand that it was limited and difficult to follow. Hence we had added several tables and figures to enhance the manuscript and make it easier for the reader to follow the numerically presented values in the results. We added a questioner to the supplemental information Figure 1 for reference. We added 1 additional figure of REBA tool, as well as two additional Tables summarizing the results of the questionnaire 

Lines 122-128 A complete description of REBA can be found in our previously published data and briefly presented here. [8] As recorded in the app, the neck, trunk and leg position were evaluated first and the total score for the evaluation provided at the end (Fig. 1A). Upper extremities were evaluated second, specifically the upper and lower arm as well as wrist position (Fig. 1B). Finally, the activity score, which accounts for the length of time in a position and the repetitiveness of activity, is given in Figure 1C. Then the final REBA score for our surgeon’s evaluation was recorded and reported in our manuscript, (Fig. 1D).

Lines 141-145 Figure 1. REBA app for intraoperative observation of ergonomic risk assessment severity. (A and B) REBA score for limbs as well as for the head and neck evaluation. (C) Activity score. (D) The REBA final risk level assessment score and recommendation, provided at the end of the assessment where the color represents severity risk as follows: green = low risk (3 or less), yellow = medium risk (4–7), orange = high risk (8–10), red = very‐high risk (11 and above).

Lines 174 Table 1. Responses to the ergonomics survey questionnaire. 

as well as 

Lines 195 Table 2. Correlation between MSK pain and years from initiating surgical training

For the specific comment Line 111-116 now reads

Lines 147-153 A chi-square test and Fisher’s exact test were used to test the association between two categorical variables. Chi-square was used when the expected number of subjects in every cell was five or more. When the expected number of subjects in any cell fell below five a Fisher's exact test was applied. To compare REBA scores between different categories of covariates a t-test was utilized. For analysis, our team used the SAS 9.4 (SAS Institute, NC) statistics software package and a p value of less than 0.05 was considered statistically significant. 

Concrete improvements:

2. Lines 49-55: Specify sources for the claim

The citations have been appropriately added within this paragraph thank you for pointing this out.

The operative tasks, which surgeons undertake everyday require not only mental sharpness, concentration, hand-eye coordination and precise execution of movement, but also minutes to hours of sustained posture with prolonged static exertion.[1, 2] While surgery is an inherently dynamic environment, where conditions change on a split-second basis, a surgeon, more often than not, assumes a poor, ergonomically-limited postural position in order to ensure that the surgical area of interest is optimally exposed and accessed. Until recently, the medical field has exclusively focused patient welfare, primum non nocere, neglecting the physician’s self-care and well-being.[3, 4]

1. Szeto GP, Cheng SW, Poon JT, Ting AC, Tsang RC, Ho P. Surgeons' static posture and movement repetitions in open and laparoscopic surgery. J Surg Res. 2012;172(1):e19-31. doi: 10.1016/j.jss.2011.08.004. PubMed PMID: 22079837.

2. Berguer R. Surgery and ergonomics. Arch Surg. 1999;134(9):1011-6. doi: 10.1001/archsurg.134.9.1011. PubMed PMID: 10487599.

3. Stucky CH, Cromwell KD, Voss RK, Chiang YJ, Woodman K, Lee JE, et al. Surgeon symptoms, strain, and selections: Systematic review and meta-analysis of surgical ergonomics. Ann Med Surg (Lond). 2018;27:1-8. doi: 10.1016/j.amsu.2017.12.013. PubMed PMID: 29511535; PubMed Central PMCID: PMCPMC5832650.

4. Voss RK, Chiang YJ, Cromwell KD, Urbauer DL, Lee JE, Cormier JN, et al. Do No Harm, Except to Ourselves? A Survey of Symptoms and Injuries in Oncologic Surgeons and Pilot Study of an Intraoperative Ergonomic Intervention. J Am Coll Surg. 2017;224(1):16-25 e1. Epub 2016/10/04. doi: 10.1016/j.jamcollsurg.2016.09.013. PubMed PMID: 27693681.

3. Line 56: Give correct definition of Ergonomics including source

The definition has been provided with it’s reference and reads as follows: 

Line 58-59 Ergonomics is “the concept of designing the working environment to fit the worker, instead of forcing the worker to fit the working environment.” [5]

5. Stylopoulos N, Rattner D. Robotics and ergonomics. The Surgical clinics of North America. 2003;83(6):1321-37. Epub 2004/01/10. doi: 10.1016/s0039-6109(03)00161-0. PubMed PMID: 14712869.

4. Line 65-78: specify further. Which studies were conducted? (including sources) What is the specific research question to be clarified in this paper? Why is it exciting?

We have referenced specific studies as per your suggestion. The specified paragraph in the introduction text now reads as follows:

Lines 67-75 While a recent meta-analysis of surgical ergonomics studies analyzed forty articles that examined subjective MSK symptoms and ergonomic outcomes through surveys across various surgical subspecialties, [3] only a few studies have looked at objective body position measurements as it relates to intraoperative ergonomics. One such study objectively assessed intraoperative ergonomics using Rapid Upper Limb Assessment (RULA) and found 0% (0/275) of pediatric otolaryngologists were found to have a negligible level of ergonomic risk. [9] Other objective tools that are commonly used to assess ergonomics include the Ovako Working Analysis System, Posture, Activity, Tools, and Handling analysis, and the Rapid Entire Body Assessment (REBA). [8]

To clarify our research question and to highlight the novelty of this study, the following has also been rewritten and edited in the draft:

Lines 84-89 This study aimed to explore the occupational risk to surgeons across multiple surgical subspecialties by comparing objective and subjective measures of ergonomic hazard. It set out to identify the prevalence of pain, prior ergonomic knowledge, and the influence of former ergonomic education on future OR behavior. Our manuscript is unique as it is the first to assess objective and subjective intraoperative ergonomic hazards and severity of MSK symptoms across multiple surgical specialties. 

5. Line 82 - 86: Unclear what 389 refers to. Did all persons participate or were they only invited?

Dear reviewer, thank you for this question, to clarify to the 389 refers to the surgeons that were invited to participate to fill out the survey. All invited personal did not participate. To further make the manuscript address this the following sentence has been added and now reads:

Lines 99-100 Of the 389 surgeons invited, 167 (43%) responded to the survey.

6. Headings: by Methods: confusing, because as a reader you do not directly understand whether you are with the theoretical background, the statistics or the results.

This paper has main headings (in capital letters) as follows: INTRODUCTION, MATERIALS AND METHODS, RESULTS, DISCUSSION, and CONCLUSION. 

Dear reviewer, thank you for this comment. We have reviewed the paper and adjusted the headings and subheadings according to the PLOS examples. We hope that this provided clarification.

We have now removed the bold of the headings and instead increased the font and then made the subheadings in bold.

we further added subsections in the Discussion section including 

Line 241 Poor knowledge of ergonomics and lack of access to ergonomically favorable furniture

Lines 267-268 Musculoskeletal discomfort as reported across surgical subspecialties and observed in the operating room

Line 305 Poor ergonomics starts early in surgeons’ career

Line 324 The current roadblocks and the future of ergonomics in surger

7. Study design and evaluation tool: What is the concrete study design? What steps were taken? What exactly was measured? How was the allocation made? Not clear

Thank you for these points. Below are sentences that were added within the methodology section to address these points and clarify the brought up questions. We measured both Subjective (via the survey) and objective (REBA intraoperative) observation assessment of ergonomic risk

Lines 93-95This was a prospective observational study of a cohort of surgeons in a tertiary hospital setting. For enrollment, our team attended Departmental meetings, where the aim of the study was explained, and surgeons were recruited at will and verbally consented for participation.

Lines 100-101 No exclusion criteria were set in place as there was an ongoing recruitment of willing participants.

Lines 131-135 The surgeons were verbally consented to participate in observation by the Ergonomics team initially prior to observation and then on the day of the observation in the operating room. Observations of the participant were conducted on the availability of the Ergonomics team as well as the surgeon on any given date. Not all of the participants that completed the survey were observed in the operating room setting.

8. relate results more strongly to the research question / hypothesis. How do the results of the subchapters help to answer the question?

Dear reviewer thank you for this insight. We hope that by clarifying our aims and objectives as modified in the introduction section and mentioned earlier addressing your initial comments, and further elaborating in the discussion as seen below, with highlighting of limitations and incorporating on the significance we have addressed this comment:

Lines 344-362 We acknowledge the limitation that certain separate aspects of our methodology have been previously examined in literature. We further recognize that although limited, the REBA methodology is used as an initial posture analysis. In order to evaluate the actual MSK load of the surgeon during the operation, tools such as surface electromyography, simulations, and other instrumented tools, would be more meaningful - especially as it relates to evaluating the underlying MSK disorders in this surgical cohort. An additional limitation is not having an interpersonal correlation, due to the anonymous nature of the survey. Thus, conclusions on an individual level between the subjective and objective measures could not be made. However, our study is unique as we tried to examine these correlations across a comprehensive list of surgical fields. Moreover, although there are limitations to our study, the ultimate benefits of intervention might only be seen in the future - after years of improved awareness and implementation of ergonomics have been adopted within the medical culture. Nevertheless, we believe there is enough evidence to support the next stage of motivating cross-institutional interventional programs in prevention and ergonomic training. In doing so, our ultimate goal is to both improve day to day complaints, and in the long run promote medical careers and improve surgeons’ quality of life. Future studies specifically looking at these parameters should be undertaken to correlate the relationship across multiple surgical fields. Overall, our results highlight the challenges as observed objectively and reported subjectively that arise from poor ergonomics in the operating room and we hope that this information will further provide insight for potential future targets for intervention. 

9. Numbers and numerical values: please check what the scientific standards are. The representation in this form is not correct.

Thank you for this comment, the values from REBA scoring were reported as a standard way based on the scores range from 0 to 15, we have adjusted within Table 4 to reflect no decimal representing the value of the score and have adjusted for the mean to now have the closest 10th decimal value. We have added additional Figure 1 to demonstrate REBA scoring system as mentioned addressing your earlier comments and as further described in our manuscript. Additionally, we have adjusted all of the percentages to be consistent across the manuscript.

10. Discussion: indicate exact studies that have already been conducted and compare your own work with them. "previous studies" is too unspecific

Dear Reviewer, thank you for this suggestion, we have adjusted our manuscript to specifically mention the highlighted cited literature by using the last name of the first author and dwelling deeper into the results of the study as it relates to ours. Below are the paragraphs that were changed

Lines 220-247 Previous studies, examining knowledge of ergonomics have found that a third or less of surveyed surgeons have any prior knowledge or training or proper OR ergonomics. A large survey study among the North American otolaryngologist residents and facial plastic surgeons showed that only 33.2% (n = 125) of participants had prior knowledge of ergonomic principles.[12] In line with this and other studies, our surgeon population had little knowledge of ergonomics (23.9%). [12-14]

Lines 274-276 This is in line with a recent meta-analysis by Stucky et al. that examined a total of 5152 surveyed surgeons looking at reported pain across multiple surgical subspecialties and finding that back (50%), neck (48%), and arm or shoulder (43%) pain was the most common. [3]

Reviewer #4: 

The paper combines three aspects of ergonomics: musculoskeletal load and risk of developing musculoskeletal disorders, knowledge regarding ergonomics and musculoskeletal pain. Idea of analysis relationship between those three is interesting and worth exploration. However, the paper teats those areas carelessly. My main objections refer to:

Authors aim of the study as: “To evaluate the relationship between presence of postural related musculoskeletal discomfort with the level of ergonomics training and the intraoperative ergonomic practice across ten surgical subspecialties” – such relationship has not been highlighted. There has been presented results in different areas.

Dear reviewer, thank you for bringing this point up. We have significantly revised our manuscript based on your and other reviewer points, with one of them, as you astutely pointed out, the incomplete/not fully explained aim. Please find our revised and more clarifying paragraph on the aim of our study and we hope that our new revision version will meet your standards for acceptable manuscript for publication.

Lines 84-89 This study aimed to explore the occupational risk to surgeons across multiple surgical subspecialties by comparing objective and subjective measures of ergonomic hazard. It set out to identify the prevalence of pain, prior ergonomic knowledge, and the influence of former ergonomic education on future OR behavior. Our manuscript is unique as it is the first to assess objective and subjective intraoperative ergonomic hazards and severity of MSK symptoms across multiple surgical specialties. 

1. There is lack of description of the tool used, especially in reference to the questionnaire. Even if it has been published, few basic informations would be helpful.

Dear Reviewer, thank you for spending time to go over our manuscript and your insight. We have added a figure demonstrating REBA scoring app as well as a Supplemental Figure with the original questionnaire to assure that it is easy for the reader to refer to the presented data. The following has also been added to the text:

Lines 77-80: The REBA is a standardized observational tool developed to enable quantitative measurement of postural strain and discomfort. It does so by scoring overall ergonomics by evaluating different body segments for muscle activity caused by static, dynamic, rapidly changing or unstable postures. [10]

Lines 122-128A complete description of REBA can be found in our previously published data and briefly presented here. [8] As recorded in the app, the neck, trunk and leg position were evaluated first and the total score for the evaluation provided at the end (Fig. 1A). Upper extremities were evaluated second, specifically the upper and lower arm as well as wrist position (Fig. 1B). Finally, the activity score, which accounts for the length of time in a position and the repetitiveness of activity, is given in Figure 1C. Then the final REBA score for our surgeon’s evaluation was recorded and reported in our manuscript (Fig. 1D).

Lines 141-145 Figure 1. REBA app for intraoperative observation of ergonomic risk assessment severity. (A and B) REBA score for limbs as well as for the head and neck evaluation. (C) Activity score. (D) The REBA final risk level assessment score and recommendation, provided at the end of the assessment where the color represents severity risk as follows: green = low risk (3 or less), yellow = medium risk (4–7), orange = high risk (8–10), red = very‐high risk (11 and above).

2. Statistical Analysis subsection informs that Chi-square test is used when the number of subjects in every cell was five or more and when the number of subjects in any cell fell below five a Fisher's exact test was applied. On what basis was accepted such rule. A chi-square is a parametric test and to apply it specific requirements must be fulfilled. It needs to be described for what purposes were those tests applied.

Thank you for this question. Fisher's exact test is always appropriate to test the association between two categorical variables, even when the sample size is big, because it is the precise way of calculating the probability based on the margins of a contingency table. When the sample size is big, it is increasingly hard to calculate the exact probability due to computing resource constraint. Pearson's chi-square test is a way to approximate Fisher's exact test. When the sample size is small, chi-square test doesn't always approximate Fisher's exact test. It is a rule of thumb to use Fisher's exact test when any expected cell value falls less than 5. We have clarified it in the following sentences in statistical method: 

Lines 147-153 A chi-square test and Fisher’s exact test were used to test the association between two categorical variables. Chi-square was used when the expected number of subjects in every cell was five or more. When the expected number of subjects in any cell fell below five a Fisher's exact test was applied. To compare REBA scores between different categories of covariates a t-test was utilized. For analysis, our team used the SAS 9.4 (SAS Institute, NC) statistics software package and a p value of less than 0.05 was considered statistically significant. 

3. There is a lot of data presented in the text. It is difficult to follow. What is the reason that Authors have not presented results of their study on figures or in tables.

Dear reviewer, thank you for this suggestion. We initially chose to present 2 tables and have since added 1 additional figure of REBA, 2 additional Tables summarizing results of the questionnaire as well as a supplemental figure to make it easy for the reader to see the questionnaire that was used in the study. 

Lines 174 Table 1. Responses to the ergonomics survey questionnaire. 

as well as 

Lines 195 Table 2. Correlation between MSK pain and years from initiating surgical training

4. What about results of statistical analysis? It is difficult to find them.

Thank you for this comment, to clarify the presentation of the results we have incorporated 2 additional tables as described above addressing your previous comment. We hope that these tables will help with visual assessment of the data in a more clear and concise manner.

5. Application of REBA requires specific procedures in three main steps: allocation of codes to each of parts of each of body posture, assessment of codes of two groups and in the and assessment of overall load. Presentation of only the final results deprives reader of interesting information. Attitude presented in this paper can be acceptable only if in Appendices are presented results of the three steps including illustration of each posture.

Thank you for this comment. The REBA scores were gathered by the colleagues from the Environmental Health & Safety Department at Stanford. We were only provided with the final score for each of the intraoperative observations and unfortunately do not have the access to individual REBA subsections, thus we cannot at this stage provide specific procedure assessment and acknowledge that this can potentially have suboptimal presentation of our results. We however hope that including additional Figures and Tables with supplemental information, created a more enhanced results section.

6. It is difficult to find on which results Authors base their conclusions.

Dear reviewer, in the conclusion section, we summarize the questionnaire's insights and the intraoperative systematic observation. Our data (based on observations in multiple departments) suggests that pain and disability resulting from poor ergonomics are widespread across surgical specialties and confirm that surgeons rarely receive implementable ergonomic training in the context of surgery (based on the survey results). Additionally, intraoperative observational findings identified that most surgeons display poor posture, particularly a poor cervical angle, which leads to increased ergonomic risk hazards and is associated with subjective cervical pain. With the low rate of ergonomic awareness, lack of access to ergonomic equipment and furniture, and high rates of MSK complaints across different stages of the surgeon’s work and outside life, the need for ergonomics education is imperative.

---

## [Decision Letter · Decision Letter 1]

26 Oct 2020

PONE-D-20-13337R1

The Risk of Ergonomic Injury Across Surgical Specialties

PLOS ONE

Dear Dr. Vaisbuch,

Thank you for submitting your manuscript to PLOS ONE. After careful consideration, we feel that it has merit but does not fully meet PLOS ONE’s publication criteria as it currently stands. Therefore, we invite you to submit a revised version of the manuscript that addresses the points raised during the review process.

We look forward to receiving your revised manuscript.

Kind regards,

Matias Noll, Ph.D

Academic Editor

PLOS ONE

Reviewers' comments:

Reviewer's Responses to Questions

**Comments to the Author**

1. If the authors have adequately addressed your comments raised in a previous round of review and you feel that this manuscript is now acceptable for publication, you may indicate that here to bypass the “Comments to the Author” section, enter your conflict of interest statement in the “Confidential to Editor” section, and submit your "Accept" recommendation.

Reviewer #1: All comments have been addressed

Reviewer #3: All comments have been addressed

Reviewer #4: All comments have been addressed

2. Is the manuscript technically sound, and do the data support the conclusions?

Reviewer #1: Yes

Reviewer #3: Yes

Reviewer #4: Yes

3. Has the statistical analysis been performed appropriately and rigorously? 

Reviewer #1: Yes

Reviewer #3: Yes

Reviewer #4: Yes

4. Have the authors made all data underlying the findings in their manuscript fully available?

Reviewer #1: Yes

Reviewer #3: Yes

Reviewer #4: Yes

5. Is the manuscript presented in an intelligible fashion and written in standard English?

Reviewer #1: Yes

Reviewer #3: Yes

Reviewer #4: Yes

6. Review Comments to the Author

Reviewer #1: I think that this manuscript adds some unique understand of the risk of ergonomic injury across surgical specialties. I thank the authors for their accurate and complete answers.

Reviewer #3: It can be seen that the authors have tried to implement all the comments, which I think is very positive. However, there are still some points that need to be improved in my opinion:

- linguistically, some improvements still need to be made. Some examples: lines 70-72 (the word "find" used several times); lines 200-229 (the word "respectivly" is used very often - finding synonyms); line 310 spelling mistakes

- Line 89: It is said that the study is unique, but this cannot be said with certainty. It may be that a comparable study has been conducted.

- Line 109: Unclear why table 1 comes pages later?

- Line 122: here, the authors are clearly referred to. Anonymity is therefore not guaranteed.

- Line 128: the reference to figure 1 is made, but you don't see the image in the text and I couldn't find it this way

- In general: locked spaces should be used to display the results. In addition, it is not always clear what the sample is that answered the corresponding questions (e.g. lines 165-170, lines 177-190)

- Line 156: unclear why eight out of ten programs were selected?

- Table 1: very confusing, because a lot has been summarized here.

- Row 275: here the only time reference is made to concrete authors by name. Inconsistent with the rest of the style

- Limitations from line 345: unclear which limitations are addressed? If the reader is not familiar with the language, he does not know what the language is about. Literature and concrete description are missing

I think if these points are implemented, it will be a really good article!

Reviewer #4: (No Response)

7. PLOS authors have the option to publish the peer review history of their article (what does this mean?). If published, this will include your full peer review and any attached files.

Reviewer #1: No

Reviewer #3: No

Reviewer #4: No

---

## [Author Response · Author response to Decision Letter 1]

12 Nov 2020

Dear PLOS Editorial and Reviewer Committee,

Thank you very much for taking the time to read our manuscript, and for your thoughtful and insightful feedback. Please see below for an itemized, point-by-point response to each of your comments. All changes in the manuscript are highlighted. We hope you find our revised version meeting the high standards of PLOS in consideration for publication.

Reviewer #1: 

1. I think that this manuscript adds some unique understand of the risk of ergonomic injury across surgical specialties. I thank the authors for their accurate and complete answers.

Dear reviewer, our authors would like to sincerely thank you for reviewing our revision manuscript again and thank you for your feedback. 

Review #3

Dear reviewer, thank you for thoroughly going over our revision manuscript and your astute suggestions. Please find a point by point response to your concerns that need addressing. We hope that this new version will meet the standard for publication in PLOS.

1. Linguistically, some improvements still need to be made. Some examples: lines 70-72 (the word "find" used several times); lines 200-229 (the word "respectivly" is used very often - finding synonyms); line 310 spelling mistakes

Thank you for pointing this out. The following changes were made within the manuscript:

Lines 70-72: One such study objectively assessed intraoperative ergonomics using Rapid Upper Limb Assessment (RULA) and observed that 0% (0/275) of pediatric otolaryngologists were found to have a negligible level of ergonomic risk.

Line 80-82 …. through observing Otolaryngologists at our institution and examining how objective and subjective scores correlated to ergonomic hazards.

We have also changed the following sentences to assure that no word is repeated multiple times within consecutive sentences:

Lines 221: Otolaryngologists (n=15, 31.3%) and ophthalmologists (n=8, 20.7%) noted experiencing discomfort in the cervical neck region while operating.

Line 235-237: The average difference in REBA scores for standing surgery were significantly higher than those for sitting (6.65 vs. 3.35, p<0.0001).

Line 301 (we could not find a grammatical error)

2. Line 89: It is said that the study is unique, but this cannot be said with certainty. It may be that a comparable study has been conducted.

Thank you for bringing this to our attention. We have changed this sentence.

Line 87-89: Our manuscript is novel as it assesses both objective and subjective intraoperative ergonomic hazards and severity of MSK symptoms across multiple surgical specialties. 

3. Line 109: Unclear why table 1 comes pages later?

Thank you for your comment. We had originally placed the survey mentioned in line 109 at the end of the manuscript because it was a supplemental table. We have now put this immediately after line 116 for clarity.

4. Line 122: here, the authors are clearly referred to. Anonymity is therefore not guaranteed.

Thank you for pointing this out, it was not our intention to de-anonymize the manuscript. We will be more careful moving forward to assure this does not happen. 

5. Line 128: the reference to figure 1 is made, but you don't see the image in the text and I couldn't find it this way

Thank you for this comment and pointing this out. The figures were separately included as requested by the journal and are available at the end of the manuscript in the pdf version. We tried to adhere to the journal preferred arrangement of the Figures (submitted as a separate file) and Tables (embedded within the text following the paragraph where it is first mentioned).

6. In general: locked spaces should be used to display the results. In addition, it is not always clear what the sample is that answered the corresponding questions (e.g. lines 165-170, lines 177-190)

Thank you for your suggestion. Given that the journal has their own standards for arranging the figures within the copy of the manuscript, we did not want to add lock spacing to our results, thus allowing the editors and the production team to be able to arrange them as they best see fit.

 In order to clarify the sample size, we have changed the following:

Lines 165-172: There were 155 participants that responded to questions on ergonomics knowledge (Table 1). Of those 155, 118 (76.1%) surgeons had no prior ergonomics training while 37 had some ergonomics training prior to the study. Eight (5.2%) stated that training occurred during medical school, 13 (8.4%) received it while in residency, 3 (1.9%) obtained it from an expert consultation, while another 13 (8.4%) reported other means of receiving training. 

In regard to ergonomic accessibility, out of 167 surgeons, 141 (84.4%) stated that they did not have access to ergonomic chair/stools in their clinic. Furthermore,120 (71.8%) surgeons reported no access to this equipment in the operating room.

7. Line 156: unclear why eight out of ten programs were selected?

Thank you for your question. We sent the survey to ten different programs, but we received no response from any surgeon from two distinct programs. We have changed the sentence to further clarify this.

Line 157-158: Out of the 389 surgeons that received the survey, 167 (42.9%) responded across eight out of ten programs that the surveys were sent to. 

 8. Table 1: very confusing, because a lot has been summarized here.

Thank you for your concern, this table was implemented during the first revision version as requested from comments by other reviewers. The data within the table corresponds to the questions that were asked in the survey (supplemental table 1). We feel that it is easy to follow, as it directly summarizes the results from the survey. Unless you strongly feel that the table should not be included, we would like to keep the table as is. Or welcome suggestions to how it might more clearly present the summarized data. 

9. Row 275: here the only time reference is made to concrete authors by name. Inconsistent with the rest of the style

Thank you for your insightful observation. We have changed this sentence and it now reads:

Line 274-277: This is in line with a recently published meta-analysis that examined a total of 5152 surveyed surgeons looking at reported pain across multiple surgical subspecialties and finding that back (50%), neck (48%), and arm or shoulder (43%) pain was the most common. [3]

10. Limitations from line 345: unclear which limitations are addressed? If the reader is not familiar with the language, he does not know what the language is about. Literature and concrete description are missing

Thank you for your comment. We have revised this line to specifically address the limitation and have cited the appropriate literature.

Line 344-345: We acknowledge that certain aspects of our study, such as subjective MSK symptoms and outcomes, have been studied in other medical specialties. [3]

Additionally, we have added citations and briefly elaborated on the suggested alternative methodologies that can be implemented as seen below: 

Line 346-352 In order to evaluate the actual MSK load of the surgeon during the operation, other tools such as surface electromyography (to be able to estimate dynamic force or fatigue during a tasks), [24] simulation-based ergonomics training curriculum, [25] and other instrumented and assessment tools, would be more meaningful - especially as it relates to evaluating the underlying MSK disorders in this surgical cohort. An additional limitation is not having an interpersonal correlation between reported symptoms on the survey and intraoperative REBA observations, due to the anonymous nature of the survey.

Reviewer #4: 

Dear reviewer, thank you for reviewing our revision manuscript again and thank you for your original feedback.

---

## [Editor Report · Decision Letter 2]

18 Dec 2020

The Risk of Ergonomic Injury Across Surgical Specialties

PONE-D-20-13337R2

Dear Dr. Vaisbuch,

We’re pleased to inform you that your manuscript has been judged scientifically suitable for publication and will be formally accepted for publication once it meets all outstanding technical requirements.

Kind regards,

Matias Noll, Ph.D

Academic Editor

PLOS ONE
---

## [Editor Report · Acceptance letter]

29 Jan 2021

PONE-D-20-13337R2 

­The Risk of Ergonomic Injury Across Surgical Specialties 

Dear Dr. Vaisbuch:

I'm pleased to inform you that your manuscript has been deemed suitable for publication in PLOS ONE. Congratulations! Your manuscript is now with our production department. 

Kind regards, 

on behalf of

Dr. Matias Noll 

Academic Editor

PLOS ONE